# Discrimination of Isointense Bitter Stimuli in a Beer Model System

**DOI:** 10.3390/nu12061560

**Published:** 2020-05-27

**Authors:** Molly J. Higgins, John E. Hayes

**Affiliations:** 1Sensory Evaluation Center, College of Agricultural Sciences, The Pennsylvania State University, University Park, PA 16802, USA; mjh91@psu.esu; 2Department of Food Science, College of Agricultural Sciences, The Pennsylvania State University, University Park, PA 16802, USA

**Keywords:** bitter, beer, discrimination, difference from control, hedonic

## Abstract

Prior work suggests humans can differentiate between bitter stimuli in water. Here, we describe three experiments that test whether beer consumers can discriminate between different bitterants in beer. In Experiment 1 (n = 51), stimuli were intensity matched; Experiments 2 and 3 were a difference from control (DFC)/check-all-that-apply (CATA) test (n = 62), and an affective test (n = 81). All used a commercial non-alcoholic beer spiked with Isolone (a hop extract), quinine sulfate dihydrate, and sucrose octaacetate (SOA). In Experiment 1, participants rated intensities on general labeled magnitude scales (gLMS), which were analyzed via ANOVA. In Experiment 2, participants rated how different samples were from a reference of Isolone on a 7-point DFC scale, and endorsed 13 attributes in a CATA task. DFC data were analyzed via ANOVA with Dunnett’s test to compare differences relative to a blind reference, and CATA data were analyzed via Cochran’s Q test. In Experiment 3, liking was assessed on labeled affective magnitude scales, and samples were also ranked. Liking was analyzed via ANOVA and rankings were analyzed with a Cochran–Mantel–Haenszel test. Experiment 1 confirmed that samples were isointense. In Experiment 2, despite being isointense, both quinine (*p* = 0.04) and SOA (*p* = 0.03) were different from Isolone, but no significant effects were found for CATA descriptors (all *p* values > 0.16). In Experiment 3, neither liking (*p* = 0.16) or ranking (*p* = 0.49) differed. Collectively, these data confirm that individuals can discriminate perceptually distinct bitter stimuli in beer, as shown previously in water, but these differences cannot be described semantically, and they do not seem to influence hedonic assessments.

## 1. Introduction

Bitterness is classically considered to be monogeusic (i.e., one singular, indistinguishable percept), and bitter stimuli that lack other side tastes are traditionally labeled with a singular semantic label in English (i.e., bitter) that lacks any additional subgroupings. However, research over the past quarter century suggests that bitterness may actually be multigeusic [1,2,3,4]. Collectively, these prior studies suggest bitter stimuli dissolved in water can be discriminated by humans. The potential ability to discriminate between bitter stimuli has important implications for the food and pharmaceutical industries, and such differences might help explain why some bitter products are more accepted (or rejected) than others. For example, learning to like one type of bitter (e.g., hops in beer) may not generalize to another type of bitter (e.g., quinine in tonic water) if the neural code for these bitters remains distinct despite a common semantic label. Here, we tested whether self-reported beer consumers could discriminate between three bitter stimuli believed to be perceptually distinct when presented within the context of a food product (here, non-alcoholic beer). After spiked beer samples were confirmed to be isointense, we tested whether participants could discriminate between the beer samples in a difference from control test. Subsequently, beer samples were also tested for affective differences.

Whether mammals can discriminate between bitter stimuli is highly controversial, as evidence for and against such an ability has been reported in humans and rats using psychophysical, behavioral, and electrophysiological data. Spector and Kopka designed a behavioral task to train rats to discriminate different taste stimuli, and at the end of training, rats were unable to discriminate quinine and denatonium [5]. This study was later extended by Martin and others, who found that rats were unable to discriminate quinine from denatonium, cycloheximide, and 6-n-propylthiouracil (PROP) [6]. Conversely, the rats showed weak discrimination of SOA from quinine. In mice, electrophysiological data shows differential responses in neural coding for a range of bitter stimuli [7]; such differential signaling would presumably be required for any behavioral discrimination.

Data from humans also presents mixed evidence of an ability to differentiate bitter stimuli. Research using semantic-free sorting and napping methods suggests that bitter stimuli can be grouped based on their perceived similarities [1,2]. Other psychophysical data in humans also suggest that bitter stimuli can be differentiated based on their temporal [3,8,9] and regional perception [4,10]. However, in a learning task designed to condition participants to associate a taste (i.e., one sweet, three bitter) with a specific color cue, participants were only able to associate the sweet sample significantly above chance; that is, there was no evidence for discrimination between the bitterants in a brief conditioning paradigm [11]. Collectively, these mixed findings suggest a need for additional research to investigate the possibility of multiple bitter percepts in humans. Further, much of the prior work has been conducted using bitter stimuli dissolved in water and some bitter stimuli have notable side tastes (e.g., bitter and astringent or bitter and salty). More work is needed to determine whether prior findings can be replicated in a more complex system that is more similar to bitter products encountered in the normal food environment. Here, we measured taste perception in humans and used a beer model system to increase the ecological validity of the data.

Humans have consumed beer for ~4 to 8 millennia [12]. More recently, the explosion of the craft beer market has increased the availability of a wide variety of beer styles, including pale ales such as IPAs and APAs (India Pale Ales, and American Pale Ales). Here, we attempted to match the flavor profile of a pale ale style beer, a style where high bitterness is accepted and even desired by consumers of these beers. Further, we recruited regular beer consumers, as they are more likely to be aware of the various flavor profiles of beer and respond positively to the bitter qualities of our samples during affective testing. Further, beer consumers’ familiarity and repeated exposure to beer makes them ideal candidates for discrimination testing. For our discrimination task, we used a difference from control (DFC) test [13,14] in lieu of more widely known discrimination methods like triangle or tetrad tests [15], given certain advantages of a DFC test. Specifically, use of a DFC test allowed us to determine whether participants could discriminate samples while also obtaining a quantitative measure for the magnitude of the perceived difference between samples. 

Here, we used both a discrimination test and an affective test; neither of these tests requires participants to generate their own verbal descriptors. The ability to accurately and descriptively discuss chemosensory perceptions often requires repeated trainings and exposure to increase recognition and recall of sensations [16]. The inability of naïve participants to describe sensations is especially pertinent to bitter taste, as bitter taste is often confused with sour taste [17] or astringency, and consumers typically lack the vocabulary to describe bitterness further without the use of hedonically loaded adjectives (e.g., “gross”, “yucky”). Still, we did add a semantic task to the DFC test in an attempt to gain additional insights on how our samples might differ. The check-all-that-apply (CATA) test is a useful tool when using untrained consumers because a CATA question provides participants with a word bank of descriptors to use to describe samples (e.g., [18]). Providing an attribute list reduces the cognitive load required of participants to generate their own descriptors.

Three experiments are reported here: an initial intensity scaling study to confirm the samples were effectively intense matched for multiple sensory attributes, a DFC test, and an affective test. Based on data from prior research on the regional [4] and temporal (Higgins, Gipple, and Hayes, under review) perception of bitter stimuli, we added quinine, SOA, or Isolone (a commercial hop extract) to beer. We hypothesized that self-reported beer consumers would be able to discriminate between the beer samples via (a) their DFC and (b) liking ratings. First, we hypothesized that Isolone and quinine would have the most different ratings in the DFC test, as these samples were most different in their regional and temporal perceptions [4]. In the affective test, we hypothesized the beer sample made with Isolone would be the most liked sample, as it is a hop extract used by the beer industry and its bitter perceptual qualities would be most similar to the bitterness normally experienced when drinking pale ale-style beers. As an exploratory measure, we also added a sensation seeking questionnaire to the affective test to explore potential relationships between personality traits and liking of the different beer samples, as other work has linked sensation seeking and the liking of a pale ale style beer (Higgins, Bakke, Hayes, in press).

## 2. Experiment One: Intensity Matching

### 2.1. Overview

Convenience samples of reportedly healthy individuals who had previously indicated interest in participating in taste and smell research were recruited on separate occasions to participate in three separate single-session (~20 min) laboratory studies. Potential participants first completed an online questionnaire. This questionnaire included eligibility screening questions, and questions about beer intake frequency. Eligible participants (described below) were invited to taste tests in a controlled laboratory setting at the Sensory Evaluation Center in the Erickson Food Science Building at Penn State. Three different studies are described here: in the first (Experiment 1), participants rated the intensity of various attributes of the beer samples; in the second (Experiment 2), participants rated how different beer samples were from a reference in a DFC task; in the final study (Experiment 3), participants provided liking/disliking ratings and a forced choice ranking of the beer samples, and also completed a personality questionnaire on sensation seeking. Data collection for all three studies occurred in semi-isolated testing booths using a computer and mouse under red light located directly overhead. Red lighting was used to minimize visual differences between the samples (see photograph in Appendix A). All data were collected using Compusense Cloud, Academic Consortium (Guelph, ONT). Participants were compensated for their time with a cash payment of $5 and gave informed consent via a click-through yes/no question on the computer screen for all studies. All procedures were approved by professional staff in the Office for Research Protections at The Pennsylvania State University (protocol #000012467).

### 2.2. Materials and Methods

#### 2.2.1. Participants

Participants (n = 55) were recruited from the main Penn State campus and surrounding community (State College, PA, USA). The participant database maintained by the Sensory Evaluation Center at Penn State contains 1400+ individuals and is composed of students, university staff, and members of the surrounding community who are age diverse (i.e., not a typical psychology study pool of undergraduates). Screening criteria included no chest cold, flu, or upper respiratory illness; currently in good health; non-smoking (i.e., no use of tobacco products in the past 30 days); not pregnant or breastfeeding; no lip/tongue/cheek piercings; no known taste or smell defects; no known allergies to quinine; no difficulty swallowing or history of choking; not under the age of 21; not taking prescription pain medication; not taking medications known to influence taste or smell function; no history of chronic pain; not abstaining from alcohol consumption for any reason (health concerns, recommendation from a health care provider, religion reasons, etc.); and willingness to consume a non-alcoholic beer sample. 

Participants were also screened based on their beer intake, and only those who reported consuming beer at least 2–3 times per month were recruited. To measure frequency of beer consumption in screening, we asked “How often do you consume beer (not including ciders, malt beverages, or spiked seltzers)” and participants indicated their intake frequency [every day, 5 to 6 days a week, 3 to 4 days a week, 2 days a week, 1 day a week, 2 to 3 days a month, 1 day a month, 3 to 11 days in the past year, 1 or 2 days in the past year, never]. The bitterness level of the non-alcoholic beer sample used here was intended to approximate the bitterness of a typical bitter pale ale style beer; thus, individuals reportedly consuming pale ales at least once per month were also recruited to participate. To avoid a demand characteristic in recruitment, lager intake was also measured. Pale ale and lager frequency of consumption were measured by asking “How often do you consume [lager-style beers (ex. Budweiser, Coors); pale ale-style beers (ex. IPAs or APAs)]?” Response options were the same as those provided for the overall beer frequency question above.

#### 2.2.2. Stimuli

The non-alcoholic beer test stimuli were made by adding various bitterants and a flavor mixture to a commercial non-alcoholic beer. We used O’Doul’s Premium (Anheuser-Busch, St. Louis, MO, USA) as the base non-alcoholic beer; it was purchased locally from the same beer distributor for all experiments. The beer came packaged in 12 oz glass bottles (~355 mL) and it was stored at room temperature prior to preparing the spiked samples. The bitter stimuli added to the non-alcoholic beer were quinine sulfate dihydrate (USP), sucrose octaacetate (FG), and Isolone^®^ (FG). As supplied by the manufacturer, Isolone is a commercial hop extract for use by the brewing industry that is a solution of 30% (*w*/*w*) iso-alpha-acid in water. These specific bitterants were chosen in light of prior work investigating temporal and regional differences of bitter stimuli [4]. Initially, we tried to use Tetralone^®^ (a hop extract of 9.5% (*w*/*w*) tetra-hydro-isoalpha-acid in water) as one of the bitterants; however, in pilot attempts to match intensity (not shown), we found Tetralone had a higher hop aroma than the other samples, which would allow participants to differentiate it from the other samples. Accordingly, Tetralone was replaced with Isolone, a similar but more concentrated extract; this allowed us to reduce odor cues while still achieving the desired amount of bitterness. 

The non-alcoholic beer was spiked with stock solutions of the three bitter stimuli: quinine (4.05 mM), SOA (2.32 mM), Isolone (1% and 0.8% of extract (*v*/*v*)). Two concentrations of the Isolone solution were prepared to increase the likelihood of matching the intensity of the Isolone-spiked beer to the quinine- and SOA-spiked beers, which had intensity matched in a previous study (data not shown). Stock solutions were made by dissolving the bitter stimuli in a 90:10 water:ethanol solution (200 proof ethanol (USP grade) in filtered water). Spiked samples were prepared by adding 5 mL of the stock and 2 mL of a flavor mixture to a room temperature bottle of beer. The flavor mixture was a 60/40 (*v*/*v*) blend of a natural grapefruit and natural orange flavoring (Brewer’s Best, Kent, OH) purchased from an online homebrewing retailer. This flavor mixture was added to help mask any additional hop aroma and flavor of the Isolone sample. After the bitter stock solutions and the flavor mixture were added to the individual beer bottles, bottles were resealed by securely twisting the cap in place. Each bottle was then inverted three times before storing under refrigeration (below 40 °F/~4.4 °C) overnight. All beer samples were prepared within 24 h of testing, and inverted again three times while capped ~30–60 min before serving. The final concentrations of the spiked beers are listed in Table 1. Final concentrations were calculated based on a final estimated 362 mL volume of beer (355 mL volume as manufactured, plus 5 mL of bitter stock and 2 mL of the flavor mixture). A control sample without any added bitterant (vehicle only) was also tested—it was prepared by adding 5 mL of a 10% (*v*/*v*) ethanol/water solution and 2 mL of the flavor mixture, to achieve the same final volume.

All samples were presented as 1.5 oz aliquots (~44 mL) in 4 oz plastic cups (~118 mL) labeled with random three-digit blinding codes. Samples were poured when participants indicated they were ready for their samples. To minimize potential differences in the samples from any separation of the added components of the bottles, participants received the same pour from each bottle (i.e., Participant 1 received only the 1st pour from all bottles, Participant 2 received only the 2nd pour, etc.). To maintain consistent carbonation and temperature, bottles were only used within 10 min of opening; any remaining beer was discarded, and fresh bottles were opened as needed. All samples were served on a single tray, evaluated one at a time, and presented in counterbalanced order using a Williams design [19].

#### 2.2.3. Procedures for Experiment One

Participants were asked to rate the intensity of multiple attributes of the beer samples. The attributes were *sweetness*, *sourness*, *bitterness*, *citrus flavor*, *other hop flavor*, and *carbonation/tingling*. Written definitions for *citrus flavor* [the flavor associated with citrus fruits such as orange, grapefruit, and lemon] and *other hop flavor* [the flavor associated with hops such as floral, fruity, herbal, etc., (excluding citrus flavor)] were provided to participants before and during ratings. Participants were instructed to take a sip of the sample, and to swallow before rating the intensity. Also, participants were instructed to rate the maximum perceived intensity of each attribute whether or not the maximum intensity was perceived while the sample was in the mouth or after swallowing. The specific instruction to swallow before rating intensity was given because prior research indicates that the maximum perceived intensity of a bitterant can occur in mouth or after swallowing, depending on the bitterant (Higgins et al., under review). An interstimulus interval (ISI) of 90 s was enforced via software after each evaluation. During this break, participants were instructed to rinse their mouths with room temperature filtered water until no sensations persisted. Spit cups with lids were provided if participants needed to expectorate during the break. 

All intensity ratings were made on a general labeled magnitude scale (gLMS). A gLMS is a semantically labeled line scale with anchors at 0 (“no sensation”), 1.4 (“barely detectable), 6 (“weak”), 17 (“moderate”), 35 (“strong”) and 51 (“very strong), 100 (“strongest sensation of any kind”) [20]. Before evaluating any samples, participants completed a scaling orientation with instructional text explaining proper scale usage and practiced using the scale by rating 15 remembered or imagined sensations [21]. Proper scale usage during the practice ratings was then evaluated using criteria from Nolden and Hayes [22]. Briefly, improper scale usage was defined as failure to correctly rank the three remembered light sensations in monotonic order, allowing for up to a 5.0 unit deviation on the 100 unit scale. Based on these criteria, four participants failed to use the scale correctly, and their ratings were excluded from subsequent analyses, resulting in a final n of 51. All other participant’s ratings in the warm-up met criteria for proper scale usage. Demographic information for the participants in all three experiments are summarized in Table 2.

#### 2.2.4. Data Analysis

All data were analyzed using SAS statistical software version 9.4 (SAS Institute, Cary, NC, USA). The Shapiro–Wilks test was used to test for normality of the intensity ratings (all *p* values < 0.001). To improve data normality, intensity ratings were square root transformed. Normality of the attribute ratings improved, but the Shapiro–Wilks test was still significant for all attributes (*p* values < 0.01). After reviewing the transformed distributions and the residual plots of the individual attribute ANOVAs, we determined that the transformations adequately improved normality, and proceeded with analysis. 

Multivariate analysis of variance (MANOVA) was performed via PROC GLM on the intensity ratings to test for an overall sample and participant effect. Following the observation of a significant sample effect using Wilk’s lambda criteria, repeated measures analysis of variance (ANOVA) were performed via PROC MIXED on each attribute using sample as a fixed effect and participant as a random effect. The Tukey–Kramer method was used to test for differences between samples following a significant F value for an individual attribute. No adjustment was made for multiple comparisons to decrease the likelihood of a Type II error. An α level of 0.05 was set for all analyses.

### 2.3. Results for Experiment One

The results of the MANOVA indicated that there was a significant sample effect [F(4, 200) = 4.29, *p* < 0.001]. The subsequent analysis of the individual attribute ratings also showed a significant sample effect for *sweetness* [F(4, 200) = 9.37, *p* < 0.001], *sourness* [F(4, 200) = 4.55, *p* = 0.002], *bitterness* [F(4, 200) = 21.89, *p* < 0.001], and *other hop flavor* [F(4, 200) = 5.52, *p* < 0.001]. The sample effect for *citrus flavor* [F(4, 200) = 0.49, *p* = 0.74] and *carbonation* [F(4, 200) = 0.11, *p* = 0.98] was not significant. Mean attribute ratings and sample comparisons are shown in Figure 1. Overall, the high concentration of Isolone sample was rated significantly higher in *bitterness* and *other hop flavor*. Upon this observation, the high concentration of Isolone (0.014%) was dropped from further testing. The lower concentration of Isolone (0.011%) was equi-intense with the quinine and SOA beers for all attributes, so these samples and concentrations were used for Experiments 2 and 3. As would be expected, the control sample containing no bitterant was significantly lower in *bitterness*, *sourness*, and *other hop flavor*, but significantly higher in *sweetness*. These data confirm the influence of the added bitterants on taste and flavor attributes in the beer. 

## 3. Experiment Two: Difference from Control (DFC) with Check all that apply (CATA)

### 3.1. Materials and Methods

#### 3.1.1. Participants and Stimuli

Participants (n = 62) were recruited using the same screening criteria as Experiment 1. The reference sample used in the DFC test was the beer sample spiked with Isolone. The test samples were beer samples spiked with quinine, and SOA; a blind replicate of the reference (i.e., Isolone) was also presented as one of the unknown samples, as is common practice in a DFC test. Specifically, this blind duplicate is used as the point of comparison against which the other test samples are judged statistically. All samples were prepared in the same manner and the same concentrations as stated above in Experiment 1. 

#### 3.1.2. Study Procedures

In a DFC test, participants are asked to compare unknown samples to a reference and rate how different the unknown samples are from the reference on a 7-point scale (0 = no difference, 6 = very large difference). Here, the Isolone-spiked sample was provided as the reference, so a blind duplicate of Isolone-spiked beer was included with the SOA- and quinine-spiked beers. Statistical comparisons are then made between the two test samples and the blind reference—this accounts for the differential willingness of participants to indicate two samples are identical (i.e., give a difference rating of zero).

Samples were served under the same conditions as Experiment 1 (i.e., red lighting, 1.5 oz aliquots, bottles discarded after 10 min, etc.) with the exception of serving design. The reference sample was presented to all participants first, followed by the three unknowns (two test samples, one blind reference) presented using a counterbalanced Williams design [19]. The unknown samples were evaluated one at a time; participants retained the reference sample and were able to retaste the reference during the evaluations of the unknown samples. 

After the DFC ratings were obtained, a check-all-that-apply (CATA) question was asked to further characterize the samples. Participants were instructed to take a sip of the sample and check all of the flavor and mouthfeel attributes that applied to the sample. Attributes from Experiment 1 were included as CATA terms and additional terms were generated via an initial pilot tasting by our team and from previous research. Specifically, terms specifying temporal percepts (e.g., slow onset, lingering aftertaste) were added based on our previous findings [4] (Higgins et al., under review). The final list of CATA items were: sweet, bitter, carbonated, hop flavor, citrus flavor, tingling, lingering aftertaste, quick onset, off flavor, sour, slow onset, quickly decreasing aftertaste, and other (with an open-ended comment box). For the CATA ratings, we emphasized temporal aspects, as we hypothesized that Isolone would be characterized by lingering aftertaste and slow onset while quinine would be characterized by quick onset and quickly decreasing aftertaste, based on other work in our laboratory. The CATA attributes were presented to each participant using a modified Williams design. The order of attributes was counterbalanced across participants but fixed within a participant; this was done to eliminate order effects while also reducing potential confusion/cognitive load that would result from randomizing the order for each sample [23,24]. 

To begin the test, participants first tasted the reference. The instructions were to take at least one sip, swallow, and make a mental note of the flavor and mouthfeel of the sample. The tasting was followed by CATA ratings for the reference. Next, participants were instructed to rinse their mouths with water during a 60 s break and wait to receive their first unknown sample. To evaluate the unknown samples, participants were instructed to taste and swallow the sample and then indicate the size of the difference between the sample and the reference sample. The difference was rated on a 7-point category scale labeled as follows: “no difference” (0), “very slight difference” (1), “slight to moderate difference” (2), “moderate difference” (3), “moderate to large difference” (4), “large difference” (5), and “very large difference” (6). Following the DFC rating for a sample, participants completed the CATA question as above. After the DFC and CATA questions were completed for each unknown, participants received a 60 s ISI where they were instructed to rinse their mouths with water before continuing to the next sample. 

#### 3.1.3. Data Analysis

The DFC ratings were analyzed using SAS statistical software. ANOVA via PROC GLM was used to test for significant differences between the ratings. Dunnett’s test was used to compare the DFC ratings of the SOA and quinine samples to the blind Isolone sample. CATA data were analyzed using XLStat version 201.9.4.1 (Addinsoft Inc., New York, NY). Cochran’s Q test was used to compare the CATA responses for each attribute for the beers spiked with SOA, quinine, and Isolone (the blind duplicate); multiple pairwise comparisons using the critical difference (Sheskin) procedure were used to test for significant differences between the samples. An α level of 0.05 was used for all analyses.

### 3.2. Results for Experiment Two

As shown in Figure 2, the mean DFC rating for the blind Isolone sample was greater than zero, suggesting that we were correct in thinking that some participants would be hesitant to use the ‘no difference’ category on the scale (i.e., an end use avoidance bias). The results of Dunnett’s test (the *p* values shown in Figure 2) indicate that the quinine- and SOA-spiked beers were both rated as being significantly different from this blind reference, indicating that both quinine and SOA can be differentiated from Isolone in a DFC task. However, the similar DFC means for quinine and SOA suggest that neither is more different from Isolone than the other.

Frequencies of the CATA attributes selected for each sample are shown in Figure 3. Cochran’s Q test using the Isolone (blind duplicate), quinine, and SOA samples found no significant differences between the attributes (all *p* values > 0.17). The Isolone (reference) sample was not included in Cochran’s test, as differences between the test samples was the primary objective of the task. Collectively, the DFC and CATA data suggest that individuals are able to differentiate some bitter stimuli in a beer model system, but that they are unable to describe the differences using a list of descriptive attributes. 

## 4. Experiment Three: Liking

### 4.1. Materials and Methods

#### 4.1.1. Participants and Stimuli

Participants (n = 81) were recruited using the same screening criteria as above. The test samples used in Experiment 3 were the beer samples spiked with quinine, SOA, and Isolone (0.011%), prepared in the same concentration and manner as Experiment 1.

#### 4.1.2. Study Procedures

Liking ratings for the spiked beer samples were made on labeled affective magnitude (LAM) scales [25]. A LAM scale is a bipolar scale with semantic affective labels [26]. The scale proceeds from left to right with the labels “greatest imaginable dislike” (−100), “dislike extremely” (−75.5), “dislike very much” (−55.5), “dislike moderately” (−31.9), “dislike slightly” (−10.6), “neither like nor dislike” (0.0), “like slightly” (11.2), “like moderately” (36.2), “like very much” (56.1), “like extremely” (74.2), and “greatest imaginable like” (100.00). Prior to evaluating the beer samples, participants rated their liking of various food items on the LAM scale to practice using the scale (e.g., [27]). The questionnaire included food and beverage items generally well liked (i.e., mozzarella cheese) and disliked (i.e., oysters) to encourage participants to use the full range of the scale. A list of all mean ratings for the food and beverage items is included in Appendix A. Participants were instructed to rate their liking/disliking of the items by clicking anywhere on the line that best represented their answer; they were told to skip an item and leave the scale blank if they had never experienced an item. 

Following the practice food and beverage questionnaire, participants evaluated the three different spiked beer samples. Participants were instructed to taste, swallow, and rate their liking/disliking after they had swallowed the sample. Participants were allowed to retaste if they wished. Samples were served as stated in Experiment 1 (i.e., red lighting, 1.5 oz aliquots, bottles opened every 10 min, served on one tray and evaluated one at a time, etc.). Between samples, participants were asked to rinse their mouths with water until no sensations persisted during a 60 s ISI. After all samples were rated, participants ranked them in a forced choice ranking (i.e., 1 = most liked, 3 = least liked). If needed, participants were instructed to retaste the samples.

Next, participants completed Arnett’s Inventory of Sensation Seeking (AISS; [28]), a 20-item replacement for Zuckerman’s Sensation Seeking Scale-V (SSS-V) [29]. Responses for individual AISS items were recorded on a 4-point category scale ranging from “does not describe me at all” (1) to “describes me very well” (4) and reverse coded when needed; these responses were summed to get an overall score. Sensation seeking is defined as “the need for varied, novel, and complex sensation and experiences, and the willingness to take physical and social risks for the sake of such experiences” [30]. Throughout this manuscript, sensation seeking (lower case) is used to indicate the underlying construct, while Sensation Seeking (capitalized) is used to indicate the summed AISS scores. A measure of sensation seeking was included here to determine whether high sensation seekers were more likely to prefer the sample spiked with Isolone. Other research on the influence of personality measures on the liking of various beer styles (Higgins et al., in press) suggests that those high in Sensation Seeking may like a bitter pale ale style beer more than those lower in Sensation Seeking. We wanted to test whether those high in Sensation Seeking might rate the Isolone sample higher in liking because this sample would be most similar to a commercial pale ale.

#### 4.1.3. Data Analysis

All data were analyzed using SAS statistical software. To test for significant differences in the liking/disliking ratings, repeated measures ANOVA was performed via PROC MIXED using sample as a fixed effect and participant as a random effect. The rank data were analyzed via PROC FREQ using the Cochran–Mantel–Haenszel test to compare the rank orders. Relationships between Sensation Seeking and liking/disliking ratings for the beer samples were analyzed using PROC CORR. An α level of 0.05 was set for all analyses.

### 4.2. Results for Experiment Three

As shown in Figure 4, ANOVA using the liking/disliking ratings of the non-alcoholic beer samples showed no significant differences in the ratings [F(2, 160) = 1.85, *p* = 0.16]. Notably, means for all samples were positive, and ranged from neither like nor dislike to like slightly. Although the ratings were not very high (in terms of widely accepted products), the samples were liked overall, not disliked. Consistent with the liking ratings, the analysis of the ranking data showed that the rank order did not significantly differ ($\chi^{2}=$ 1.41, *p* = 0.49) across samples. The frequency of the rank orders and the total sum of all rankings for each sample are shown Figure 5. All samples were ranked 1st at approximately an equal frequency. Notably, the Isolone sample was ranked 3rd most frequently, suggesting that the sample might be more polarizing than the other samples. No correlations between the Sensation Seeking scores and the liking/disliking ratings of the samples (Table 3) were observed (all *p* values > 0.91).

## 5. Discussion

The overall goal of the research described here was to determine whether self-reported beer consumers could differentiate perceptually distinct bitter stimuli (Isolone, quinine, and SOA) in a beer model system, and whether certain types of bitter stimuli were more liked or disliked. Experiment 1 demonstrated that the spiked beers were isointense for bitterness and other attributes, while in Experiment 2, the DFC data showed that individuals were able to differentiate a sample spiked with Isolone from samples spiked with quinine and SOA. However, the non-significant results from the CATA data suggest that consumers, as a group, were unable to describe the differences between the samples, even when provided a list of attributes. In the affective tests in Experiment 3, the absence of significant differences for liking ratings or rank orders suggests that, as a group, the self-reported beer consumers tested here equally liked all beer samples and did not differentiate the samples in terms of liking or preference.

Our original hypotheses were only partially supported by present data. Namely, quinine and SOA samples were differentiated from Isolone in the DFC test, even when intensity matched. Reasons for the inability to support the remaining hypotheses are unknown, but may include our use of a DFC test rather than more comprehensive and exhaustive methods like descriptive analysis, or use of a beer matrix where additional ingredients can mask subtle differences in bitter taste perception. The premise of this research was based on evidence from previous research demonstrating that humans can discriminate bitter stimuli in water based on differences in bitter percepts [1,2,4] (Higgins et al., under review). Notably, all this previous work was conducted in water, and we cannot assume that all perceptual qualities of bitterants in water would be salient in a more complex beverage matrix like non-alcoholic beer. While ethanol greatly enhances and contributes to the flavor profile of beer [31,32,33], non-alcoholic beer remains a complex matrix, containing iso-alpha-acids, glycerol, carbonation, and flavor compounds which contribute to its taste, aroma, and mouthfeel. Knowing this, it is possible that certain perceptual qualities of the bitter stimuli were masked or altered in our beer model system. Further, the DFC test did not detect differences between the beers spiked with SOA and quinine because the Isolone-spiked beer was the reference. That is, the focus of the task was to determine whether the samples could be differentiated from the Isolone reference sample, not each other. Further, the results of the affective test can only demonstrate whether differences in liking exist between the beer samples, not whether or not the samples are perceptibly different. Collectively, it seems our participants were able to perceive a difference between the samples in Experiment 2, but based on our data in Experiment 3, these perceptual differences did not affect liking or preference. 

The failure to observe significant differences for the liking of the beer samples may stem from the underlying process through which individuals learn to tolerate or like bitter-tasting products. Humans are born with the innate disliking of bitterness [34], but this initial aversion may be overcome through a variety of potential means, such as flavor-consequence learning (FCL) [35]. FCL occurs when a flavor in a food or beverage (the conditioned stimulus, e.g., coffee flavor and aroma) is paired with a positive post-ingestive effect (the unconditioned stimulus, e.g., caffeine) and the repeated pairing of the conditioned stimulus and the unconditioned stimulus in the food of beverage leads to a modification in the affective response to the conditioned stimulus. However, once liking or acceptance is established for a product, liking does not necessarily generalize to other products with similar taste qualities. For example, in a study where participants repeatedly consumed a sweet/bitter beverage (SanBitter), affective ratings increased for the beverage, but not for other products with bitter and sweet tastes [36]. We originally hypothesized that this relationship may be due to different bitter percepts found in the products. For example, the lingering, back-of-mouth bitterness from the iso-alpha-acids in pale ales may prevent gin-and-tonic drinkers from becoming pale ale consumers if tonic drinkers are familiar with and prefer the quick onset, quick decaying bitterness of quinine. However, the failure to observe significant differences in liking ratings here in Experiment 3 led us to reconsider our original hypothesis. Accordingly, we now speculate that when individuals are repeatedly exposed to bitter food or beverage products, any learned liking occurs for the product or the product category, rather than the unique bitter percepts found within that product. Under this revised hypothesis, we can presume participants in Experiment 3 (i.e., pale ale consumers) were previously conditioned to like bitterness within the context of pale ales, and not the specific bitterness from the hop extracts. Accordingly, replacing the bitterness from hops with the bitterness of quinine or SOA did not meaningfully affect liking because the subtly different bitter subqualities (if they exist) all occurred in the same beverage context. More work is needed to confirm this revised hypothesis. 

Additionally, previous data indicate that craft beer consumers generally report higher overall liking of beer [37,38], and it is possible that the participants in Experiment 3 would report equal liking for all beer samples, regardless of bitter quality or beer style (e.g., pale ale, lager). Because of this, we added the forced choice rank question in Experiment 3. However, the results of the forced choice rankings aligned fully with rated liking and no significant differences were observed, providing further evidence that beer consumers generally like most beers.

Our exploratory hypothesis regarding the effect of sensation seeking on the liking of the beer samples was also not supported. Other data suggest that personality traits can influence oral burn, an innately disliked sensation [39,40]. Because bitter is also innately disliked, we reasoned that we would observe a similar relationship here, where individuals high in Sensation Seeking would also report higher liking ratings for the beer samples. This relationship was observed in a separate study, where liking ratings for a bitter pale ale beer was positively correlated with Sensation Seeking (Higgins, et al., in press). Here, however, we did not observe any significant correlations between Sensation Seeking and total liking of all beers or liking of an individual beer. Reasons behind the absence of a relationship here are unknown, but may be due to the isointense bitterness of all beers; that is, variation in bitterness intensity across products may be needed to observe a relationship with sensation seeking, and we eliminated any such variation with our intensity matching procedure. 

Additionally, our findings demonstrate mixed support of previous research studies comparing the perceptions of multiple bitter stimuli. In rats, Martin and others noted a weak discrimination between quinine and SOA [6] and in humans, Paravisini and colleagues noted similar placement of the two stimuli in a napping experiment [2]. Conversely, when using data from a sorting task in humans, analysis by McDowell placed quinine, SOA, and a hop extract into three separate groups [1], and Higgins and Hayes noted that SOA and a hop extract shared more regional and temporal qualities than quinine [4]. Notably, our conclusions here regarding comparisons between SOA and quinine are limited because all rated differences were made in comparison to the Isolone sample. A new DFC test with quinine or SOA as the reference would be necessary to compare quinine and SOA. Still, our findings show that a beer spiked with Isolone, a hop extract, was perceivable different from those made with quinine and SOA, partially supporting the findings of McDowell, and Higgins and Hayes. Further, discrepancies with prior data may have occurred because the prior research used bitter stimuli dissolved in water, not a complex matrix like beer. Additionally, the sorting studies used distinct methods (i.e., sorting versus sorted napping; see [41] for an overview of the methods) and a distinctive set of bitter stimuli which affects the placement and groupings of stimuli and, ultimately, the overall comparisons and findings. For example, Paravisini and others used bitterants with additional side tastes and non-taste qualities—magnesium sulfate is salty [42], while catechin and epicatechin are astringent [43]. These other qualities may have been major drivers of their placement and groupings, as the addition of side tastes alters the placement of stimuli on a perceptual map [44]. In contrast, McDowell [1] only used bitter stimuli with minimal or no side tastes. 

The napping experiment by Paravisini and others served as preliminary data for a subsequent study on the influence of bitterness on retronasal aroma in a coffee model system [2]. In their data, they found the perceived aroma of a coffee isolate varied based on the added bitterant (e.g., the coffee isolate paired with caffeine and catechin had a higher perceived hazelnut aroma than coffee isolates paired with quinine and L-tryptophan). Those authors speculated that the perceived differences were caused by the cognitively driven cross-modal interaction between taste and aroma. This hypothesis is also supported by work involving hop extracts, where increased levels of hop aroma extract altered the perceived bitterness and bitter character (i.e., harsh versus round) of beer [45]. While we have no direct evidence on the interaction between our bitter stimuli and the flavor constituents of our beer samples, it seems reasonable to suspect this phenomenon may have contributed to the perceived differences between our beer samples. Notably, the Isolone hop extract used here has an aroma. However, the added citrus flavors were used to help mask the aroma, and the intensity ratings from Experiment 1 show no evidence that the Isolone sample had a higher perceived hop aroma. Additional methods, such as descriptive analysis [46,47] could be used to confirm the intensity ratings using a trained panel, versus the naïve consumers used here. Further, descriptive analysis methods might also determine whether there is evidence of cross-modal interactions between taste and aroma in the beer samples used here. However, findings using descriptive analysis methods may be limited as prior research suggests that analytical evaluation methods (i.e., the separation of taste and flavor components of a product [48]) minimize or eliminate cross-modal interactions shown using synthetic (i.e., holistic) evaluation methods [49]. 

Some limitations of the current research should also be noted. One issue with the approach used here was our choice to intensity match the bitterants at a group level rather than at an individual level. Bitter taste perception can vary between individuals due to polymorphisms in *TAS2R* bitter taste genes that influence receptor function [50,51,52]. If some individuals were genetically predisposed to perceive various bitter stimuli at different intensities, differences in bitter intensity (rather than differences in bitter quality) might conceivably explain the significant DFC ratings found in Experiment 2. However, for multiple reasons, we do not believe genetic differences in bitterness can explain our results and interpretation. First, we drew participants for all three studies from the same general population, so there is no reason to suspect the proportion of functional alleles will vary systematically across the three studies (as might occur with studies at different locations drawn from different populations). Second, the number of participants in each study are greater than numbers classically used in many psychophysical experiments; critically, with a larger sample size, the potential for an atypical random sample that has a genetic makeup which differs idiosyncratically from the population the sample is drawn from decreases substantially, both because sampling bias drops, but also because any potential influence of a few atypical genotypes on overall group performance drops as well. Further, two of the three compounds used in our study bind to TAS2R receptors with no known functional variation. Specifically, the isohumolones found in hops ligate TAS2R1 and TAS2R14, and sucrose octacetate (SOA) ligates TAS2R46: none of the genes that encode these receptors are known to contain functional polymorphisms. Accordingly, the discrimination of SOA from Isolone in Experiment 2 cannot be attributed to genetic variation (also see [53]). Still, we should note that quinine, the third compound used here, does show some functional variation with polymorphisms in *TAS2R31* [52,54]. Given the other points above, however, we do not believe this variation meaningfully influences our conclusions, as it cannot speak to the difference between SOA and Isolone. Finally, other evidence from humans suggests that even when clear intensity differences are present, bitterants are still sorted into the same group based on qualitative differences. Specifically, McDowell [1] found high and low concentrations of Tryptophan clustered together in free sorting, despite having clearly different intensities. Thus, while we cannot completely rule out any genetic effects (as participants in the present experiments were not genotyped), we do not believe that such functional differences would be able to systematically bias the results reported here. 

Further, we were able to observe significant differences in sweetness, bitterness, sourness, and other hop flavor for the Isolone (high) and control (no bitterant) samples (Figure 1), indicating that our sample cohort was able to detect differences in intensity of samples of distinct concentrations. Further, the CATA data from Experiment 2 showed no significant differences for the bitter attribute. Nevertheless, an alternative approach to reduce the effect of individual variation in bitter taste perception would be to intensity match all compounds for each participant. However, this approach would be extremely resource intensive in terms of experimenter time, and places a heavy burden on participants—both of which would reduce the number of participants that could be tested, which would then reduce the power of the statistical tests. Accordingly, we decided against this approach, but cannot rule out the possibility that doing so might provide different results.

Additional limitations of our research include use of naïve consumers rather than trained participants. We used untrained consumers in Experiment 3 as classical dogma in sensory and consumer science indicates liking tests should not be performed with trained panelists [55]. Further, we used untrained consumers for the DFC task, as training does not affect the ability to discriminate beers: previous research has shown that trained and untrained beer consumers perform equally when asked to sort [56,57,58] or discriminate unfamiliar beers [59,60]. Nevertheless, our use of naïve consumers may limit utility of the CATA ratings in Experiment 2, as prior work shows that terms generated by untrained panelists to describe beer lack consensus and are less specific than those of trained panelists [57,60]. Notably, we provided a list of pre-generated terms to our participants. In other work, providing a long list of terms (i.e., [44]) did not improve trained or untrained panelist’s ability to later match beers to their selected descriptors [58]. Conversely, in wine, when untrained panelists are provided with a shorter list of descriptors (i.e., 14 terms), matching performance improves [61]. Here, our CATA list had 13 attributes, which presumably lessened effects of using untrained consumers. Still, some participants may not be familiar with all attributes in our CATA list, so our non-significant CATA findings may be due to a lack of consensus in the meaning of terms. Further, additional relevant CATA attributes, such as chemesthetic sensations, may have been overlooked during creation of the attribute list. Collectively, these limitations could be reduced by using a trained descriptive panel, as the attribute trainings and term generation exercises in descriptive analysis might allow for deeper characterizations and descriptions of the samples, thereby reducing error. 

Finally, while our current model is more complex than simple aqueous solutions, it remains less multifaceted than solid foods, and additional variables such as mastication may also play a role in the discrimination of bitter perceptual qualities, given the relationship between chewing and salivary flow.

## 6. Conclusions

Our results provide some evidence to support the notion that humans are able to discriminate between bitter stimuli. Self-reported beer consumers were able to distinguish a beer sample containing Isolone, a hop extract, from beer samples containing quinine or SOA. However, we were unable to identify semantic labels for the perceptual qualities of these stimuli, as no attributes from the CATA data were significant. In affective testing, we did not observe any significant differences in the liking ratings or rank orders for the beer samples, suggesting the perceptual qualities of the specific bitter stimuli did not influence liking. This led to the generation of an alternative hypothesis—when learned liking for bitter food and beverage products occurs, liking develops for the product or the product category, and not the unique bitter percepts of those bitterants. Additional research is needed to test this hypothesis and to identify the perceptual qualities of the beer samples tested here; more generally, additional work is also needed to explore how the perceptual qualities of bitter stimuli influence flavor perception of other complex food and beverages.

## Figures and Tables

**Figure 1 nutrients-12-01560-f001:**
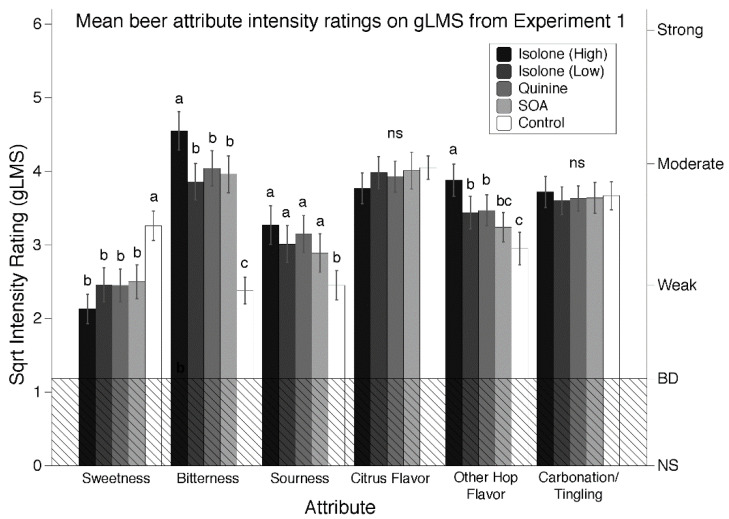
Square root transformed mean attribute intensity ratings and SEM made on a general labeled magnitude scale (gLMS). The hatched region at the bottom of the figure indicates ratings below barely detectable. The letters above the intensity ratings indicate the Tukey groupings for the least square means (lsmeans) comparisons. Samples that do not share a letter are significantly different via Tukey’s HSD at *p* < 0.05. Attributes with no significant sample effect are noted using “ns” above the ratings. No adjustments for multiple comparisons were made.

**Figure 2 nutrients-12-01560-f002:**
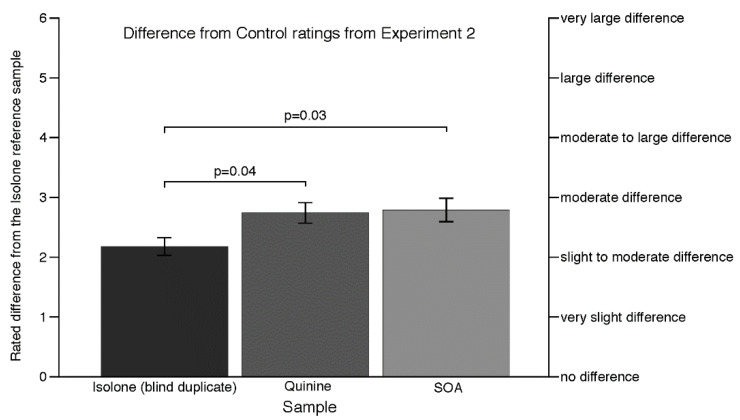
Rated differences (mean and SEM) of the test samples from the Isolone reference sample in Experiment 2. Comparisons between samples are from Dunnett’s test using the Isolone (the blind reference) sample as a control. Higher ratings indicate a larger difference from the reference sample while lower ratings indicate a smaller difference from (i.e., more similarity with) the reference sample.

**Figure 3 nutrients-12-01560-f003:**
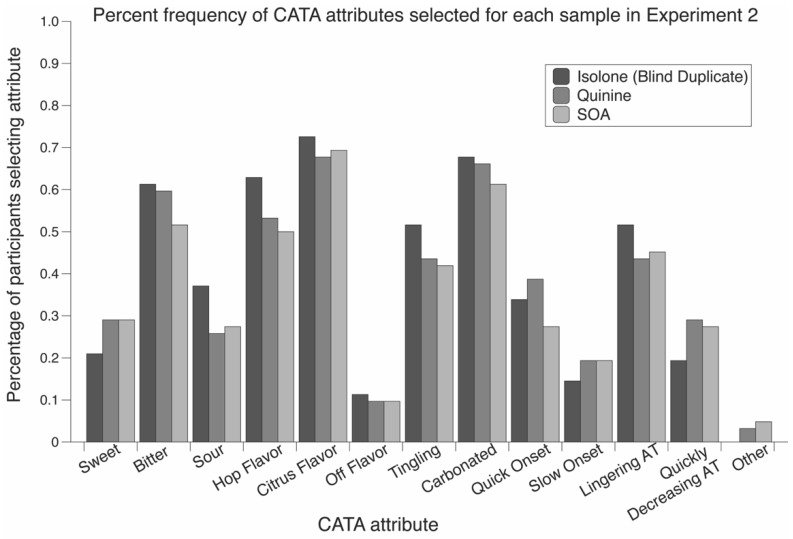
Percent frequency of check-all-that-apply (CATA) attributes selected for each sample in Experiment 2. Cochran’s Q test to test for significant differences was run using the unknown samples: Isolone (blind reference), quinine, SOA. No significant differences were observed for the CATA attributes (all *p* values > 0.17). Aftertaste is abbreviated as AT.

**Figure 4 nutrients-12-01560-f004:**
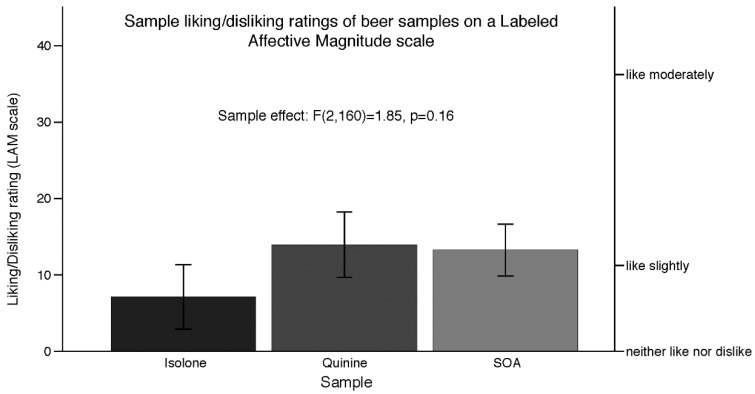
Mean liking/disliking ratings and SEM of the beer samples on labeled affective magnitude (LAM) scales from Experiment 3. No significant differences between the samples were observed.

**Figure 5 nutrients-12-01560-f005:**
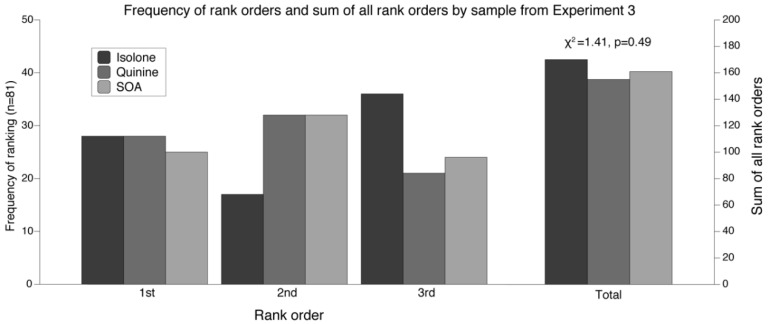
Frequency of rank orders and sum of rank orders by sample from Experiment 3. No significant differences between sample total rank orders were observed. Note the left y-axis is for the rank order frequencies and the right y-axis on is for the sum of all rank orders.

**Table 1 nutrients-12-01560-t001:** Bitterants used to spike beer samples in Experiments 1, 2, 3. Note: the high Isolone concentration was only used in Experiment 1.

Stimulus	Final Concentration	Source
Quinine sulfate dihydrate	0.056 mM	Fisher Scientific, Fair Lawn, NJ
Sucrose octaacetate	0.032 mM	SAFC Supply Solutions, St. Louis, MO
Isolone (High)	0.014% (*v*/*v*)	Kalsec, Kalamazoo, MI
Isolone (Low)	0.011% (*v*/*v*)	Kalsec, Kalamazoo, MI

**Table 2 nutrients-12-01560-t002:** Demographic information for participants from all experiments.

Experiment	n	Age (Mean, ± SD)	Frequency Group	Beer Intake (n)	Pale Ale Intake (n)	AISS (Mean, ± SD)
1	n = 51 (20 men, 31 women)	35.0 ± 11.3	Weekly	40	25	-
			Monthly	9	21	
			Yearly	2	5	
2	n = 62 (20 men, 42 women)	37.3 ± 13.4	Weekly	48	32	-
			Monthly	14	27	
			Yearly	0	3	
3	n = 81 (35 men, 46 women)	38.0 ± 12.8	Weekly	64	42	Total: 50.2 ± 7.3
			Monthly	16	34	Men: 52.8 ± 7.6
			Yearly	1	5	Women: 48.3 ± 6.5

(-) indicates not measured.

**Table 3 nutrients-12-01560-t003:** Pearson correlation coefficients of liking/disliking ratings of the non-alcoholic beer samples, demographics, and Sensation Seeking. Bolded values indicate significant correlations.

	Quinine	SOA	Total Liking	AISS	Age	Sex
Isolone	**0.56** ***	**0.48** ***	**0.81** ***	−0.01	−0.06	−0.04
Quinine		**0.63** ***	**0.87** ***	0.00	0.10	−0.12
SOA			**0.83** ***	−0.01	0.10	−0.17
Total Liking				−0.01	0.05	−0.13
AISS					−0.20	**+0.31** **
Age						−0.17

** *p* < 0.01, *** *p* < 0.001; for the sex variable, women were coded as 0 and men were coded as 1; total liking is the sum of all liking/disliking ratings for each participant.

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
