# Peer review of "Discrimination of Isointense Bitter Stimuli in a Beer Model System"

_nutrients, 2020, doi:10.3390/nu12061560_

Round 1

Reviewer 1 Report

Two experimental points limit the author’s ability to draw the conclusions they do about the ability of subjects to discriminate bitter compounds when those compounds are added to the beer. While the attempt to address these points in the Discussion section is appreciated, unfortunately, if the intensity matching was done on a different group of people than those who participated in Experiment 2, the differences in intensity among the three beers (each spiked with a different bitterant) could be wholly or partly due to differences in the individual differences in inborn bitter sensitivity among a new group of subjects. While the authors can conclude that the averages intensity rating of the groups differs in Experiment 2, they cannot claim this is because the isointense bitter stimuli can be discriminated. A second and equally important point is that the use of food-grade compounds is understandable from a human safety point of view, but impurities including odorants could account for the ability to discriminate the beer samples. This is a vexing problem in human research but unfortunately limits the ability to make claims the authors wish to make about the data. For example, odorants from Isolone might be differentially perceived by subjects in Experiment 2 and these differences may wholly or partly account for the differences in that stimuli feature. Olfactory researchers are realizing that the use of impure odorants has hampered the ability to better understand odor coding and while difficult to deal with experimentally, using single-compound stimuli ideally verified by chemical analysis methods is essential.

Author Response

We thank both reviewers for their input. Based on the feedback, we have revised the manuscript to include further justifications and clarifications of the approaches used here. We have also tried to better clarify the limitations of these studies.

Reviewer 1

Two experimental points limit the author’s ability to draw the conclusions they do about the ability of subjects to discriminate bitter compounds when those compounds are added to the beer. While the attempt to address these points in the Discussion section is appreciated, unfortunately, if the intensity matching was done on a different group of people than those who participated in Experiment 2, the differences in intensity among the three beers (each spiked with a different bitterant) could be wholly or partly due to differences in the individual differences in inborn bitter sensitivity among a new group of subjects. While the authors can conclude that the averages intensity rating of the groups differs in Experiment 2, they cannot claim this is because the isointense bitter stimuli can be discriminated.

Reply: Thank you for this important comment. We apologize for not better addressing this understandable concern in our original submission. While we concede that genetic differences in bitterness are certainly well documented and are important (e.g., Hayes et al. 2015 Chem Senses; Hayes et al. 2008 Chem Senses), we do not believe they can meaningfully influence our results and interpretation here for several reasons.

Foremost, we drew participants for each study from the same general population, so there is no reason to suspect genetics will vary systematically across the three studies (as might occur with studies at different locations drawn from different populations). Second, the number of participants per study are relatively large for a psychophysical study of this type (i.e., bigger than classical n’s close to 25 or 30), and with a larger sample, the potential for an atypical random sample that has a genetic makeup that differs idiosyncratically from the sample it is drawn from decreases substantially, both because representativeness increases, but also because any potential influence of a few atypical genotypes on overall group performance drops. Third, even if we ignore these first two points, two of the three compounds used here bind to TAS2R receptors with no known functional variation. Specifically, the isohumolones found in hops ligate TAS2R1 and TAS2R14 (http://bitterdb.agri.huji.ac.il/), and neither of these genes contain functional polymorphisms. Likewise, sucrose octacetate (SOA) ligates TAS2R46, which does not contain any functional polymorphisms. Conversely, we should acknowledge that the third compound used here, quinine, does show some functional variation with TAS2R31 SNPs (e.g., Hayes et al. 2015); still, given points one and two above, we do not believe such variation would meaningfully influence our results. Finally, other evidence from humans suggests that even when clear intensity differences are present, bitterants are still sorted into the same group based on qualitative differences – specifically, McDowell (2017) found that high and low concentrations of Tryptophan still clustered together in free sorting, despite having different intensities. The discussion in the revised manuscript now details these points more clearly.

A second and equally important point is that the use of food-grade compounds is understandable from a human safety point of view, but impurities including odorants could account for the ability to discriminate the beer samples. This is a vexing problem in human research but unfortunately limits the ability to make claims the authors wish to make about the data. For example, odorants from Isolone might be differentially perceived by subjects in Experiment 2 and these differences may wholly or partly account for the differences in that stimuli feature. Olfactory researchers are realizing that the use of impure odorants has hampered the ability to better understand odor coding and while difficult to deal with experimentally, using single-compound stimuli ideally verified by chemical analysis methods is essential.

Reply: Thank you for this comment. We feel this reflects a classic and unresolvable trade off between the added experimental control that comes from using purified single compounds in DI water versus the ecological validity that comes from testing real foods and natural products. We feel both are important: compare Higgins and Hayes 2019 Chem Senses, which uses pure compounds, to the present work which uses real foods.

Reviewer 2 Report

The study is well designed and well conducted congratulations.

However, there is one point that needs to be carefully elucidated.

In the text at line 336 and in the legend of Figure 3, at lines 343-346, the Authors state that the Cochran’s Q test to test for significant differences was only run using the unknown samples: Isolone (blind reference), quinine, SOA. However, in the graph, they also inserted the column for Isolone reference. This makes it confusing for the reader.

Moreover, the authors state that “the Isolone (reference) sample was not included in the Cochran’s test as differences between the test samples was the primary objective of the task.” However, in the graph the column of Isolone (reference) and Isolone (blind reference) seem different for bitter, citrus flavor and tingling. Did the authors perform any statistical analyses to test for significant difference between these two samples?

Line 343: Please correct the number of figure: it should be Figure 3 not Figure 1.

Author Response

We thank both reviewers for their input. Based on the feedback, we have revised the manuscript to include further justifications and clarifications of the approaches used here. We have also tried to better clarify the limitations of these studies.

Reviewer 2

The study is well designed and well conducted congratulations.

However, there is one point that needs to be carefully elucidated.

In the text at line 336 and in the legend of Figure 3, at lines 343-346, the Authors state that the Cochran’s Q test to test for significant differences was only run using the unknown samples: Isolone (blind reference), quinine, SOA. However, in the graph, they also inserted the column for Isolone reference. This makes it confusing for the reader.

Reply: We revised Figure 3 by removing the Isolone reference to reduce confusion to the readers.

Moreover, the authors state that “the Isolone (reference) sample was not included in the Cochran’s test as differences between the test samples was the primary objective of the task.” However, in the graph the column of Isolone (reference) and Isolone (blind reference) seem different for bitter, citrus flavor and tingling. Did the authors perform any statistical analyses to test for significant difference between these two samples?

Reply: We did perform an additional analysis on the CATA responses with the Isolone (reference) sample and did not find significant differences for certain attributes. However, we did not include this finding in the ms due to the sample evaluation method used here (i.e., the reference was always presented first and was potentially subject to order effects, participants were primed to expect a difference in the test samples).

Line 343: Please correct the number of figure: it should be Figure 3 not Figure 1.

Reply: Figure number was corrected.

Reviewer 3 Report

Please change units to internationa system (temperature and volume)

Author Response

C and mL have been added where appropriate.

Round 2

Reviewer 1 Report

All concerns raised during the initial review have been adequately addressed in this revision.